# Unsupervised Reconstruction of Sea Surface Currents from AIS Maritime Traffic Data Using Trainable Variational Models

**Simon Benaïchouche** [1,2,*], **Clément Legoff** [2], **Yann Guichoux** [2], **François Rousseau** [3] and **Ronan Fablet** [1]

1    IMT Atlantique, UMR CNRS Lab-STICC, CEDEX 2, 29238 Brest, France; ronan.fablet@imt-atlantique.fr
2    Eodyn, 29280 Plouzané, France; clement.legoff@e-odyn.com (C.L.); yann.guichoux@e-odyn.com (Y.G.)
3    IMT Atlantique, UMR INSERM LaTIM, CEDEX 2, 29238 Brest, France; francois.rousseau@imt-atlantique.fr
*    Correspondence: simon.benaichouche@imt-atlantique.fr

**Abstract:** The estimation of ocean dynamics is a key challenge for applications ranging from climate modeling to ship routing. State-of-the-art methods relying on satellite-derived altimetry data can hardly resolve spatial scales below ∼100 km. In this work we investigate the relevance of AIS data streams as a new mean for the estimation of the surface current velocities. Using a physics-informed observation model, we propose to solve the associated the ill-posed inverse problem using a trainable variational formulation. The latter exploits variational auto-encoders coupled with neural ODE to represent sea surface dynamics. We report numerical experiments on a real AIS dataset off South Africa in a highly dynamical ocean region. They support the relevance of the proposed learning-based AIS-driven approach to significantly improve the reconstruction of sea surface currents compared with state-of-the-art methods, including altimetry-based ones.

**Keywords:** AIS; inverse problems; data assimilation; deep learning; neural ODEs





## 1. Introduction

Over the last decades, in a context of globalization, we have observed an exponential growth of maritime traffic. To avoid ship collisions and improve safety at sea, the International Maritime Organization (IMO) mandated in 2004 the use of the Automatic Identification System (AIS). Nowadays AIS data streams provide hundreds of millions of messages every day [1]. In the era of Big Data and as suggested in [2,3], this massive amount of information might complement remote sensing technologies for ocean observation purposes.

Until the late 1980s, the estimation of current velocity relied on the logs of merchant and military ships [4,5]. Such data provided a very scarce observation of sea surface dynamics. Satellite radar altimetry missions has been revolutionizing the observation, modeling and understanding of ocean dynamics on a global scale for 30 years. They have greatly participated to revealing the importance of the so-called ocean mesoscale and submesoscale dynamics, which typically refer to horizontal scales from a few hundreds of meters to hundreds of kilometers. While satellite altimetry data cannot solve scales below ∼100 km due to narrow-swath sampling patterns, other technologies have been developed such as drifters, high-frequency (HF) radar technologies or SAR imaging [6]. Their space–time sampling however involve some limitations to fully inform sea surface dynamics on a global or even regional scale.

Following [2,3], this work further explores the potential of AIS data streams to infer sea surface dynamics. The irregular space–time sampling of AIS data, the complex relationships between ship trajectories and sea surface conditions as well as the absence of a consistent ground-truth dataset of fully observed fields make this objective particularly challenging. Here, we develop a novel learning-based scheme, which relies on a variational formulation of the underlying inverse problem. This approach allows us to exploit some a

priori knowledge on the considered problem along with the computational power of deep learning approaches. Our key contributions are as follows:

- Drawing inspiration from a 4D-Var data assimilation formulation [7], we cast the estimation of sea surface current from AIS data streams as a minimization problem that involves a physics-informed observation term coupled with a trainable ordinary differential equation (ODE) prior. This formulation relates to variational auto-encoders (VAEs) and exploits external data to regularize the considered ill-posed inverse problem.
- The proposed learning scheme applies directly to AIS data streams with no requirement for a groundtruthed dataset for sea surface currents. As such, it is regarded as a non-supervised approach.
- Being implemented with a deep learning framework, namely Pytorch, we benefit from GPU acceleration to process AIS data streams, which is critical for scaling up to global AIS datasets.
- Numerical experiments for a real dataset demonstrate the relevance of the proposed approach with respect to state-of-the-art approaches. As case-study region, we focus on the Aghulas current off South Africa. This region involves challenging and complex sea surface dynamics for altimetry products. We report significant improvements up to 40% w.r.t. state-of-the-art altimetry-derived assimilation-based products [8,9].

The paper is organized as follows. Section 2 briefly reviews related studies and presents the considered observation model. We describe the proposed learning-based approach in Section 3. In Section 4 we report numerical experiment and benchmark with state-of-the-art product on a real case-study and Section 5 provides concluding remarks.

## 2. Problem Statement and Related Work

### 2.1. Related Work

AIS data streams contain features such as a ship identifier denoted as a maritime mobile service identity (MMSI), the time of emission, GPS speed over ground (SOG), course over ground (COG) and positions in latitude and longitude coordinate (lat,lon). AIS data also include the true heading of the ship along with static information (e.g., draught, width and length). Over the past few years, many studies have investigated the automatic analysis of maritime data using satellite aperture radar (SAR) [10,11] or AIS for the surveillance of the maritime traffic [12,13]. In [13] deep-leaning-based techniques are shown to provide a relevant framework to deal with key features of AIS data such as noisy and corrupted messages and an irregular time sampling. Given the fact that the motion and trajectory of ships may be affected by the sea surface state, a few studies [2,3] have investigated whether one might explore AIS data to monitor ocean surface conditions. In [3], the focus is given on the monitoring of a tsunami event, whereas [2] addresses the reconstruction of sea surface velocity fields using an heuristic method.

Following this line of work, we investigate an unsupervised deep-learning-based approach to unravel the sea surface current conditions encoded in AIS data streams.

### 2.2. Observational Model

Under ideal conditions, that is to say with no external forcing due to strong wind or waves, the motion of a ship as pictured in Figure 1 can be modeled using the following linear equation:

$$
\begin{aligned}
S_{ground}(t) &= L(U(p,t), S_{surf}(t), H(t)) \\
&= S_{surf}(t)H(t) + U(p,t)
\end{aligned}
\tag{1}
$$

where $U(p,t)$ denotes the sea surface current at a given ship location $p$ at time $t$, $S_{ground}(t)$ refers to the speed over ground, $S_{surf}(t)$ to the surface speed of the boat, $H(t) \in S^1$ a vector pointing towards the known heading direction of the ship. With a single observation we cannot invert the linear system described by Equation (1). However, assuming that sea

surface current conditions evolve slowly in space and time and that the density of ships is large enough, we may derive a least-square solution as explored in [14]. The performance of this approach is however highly dependent of the conditioning of the linear system as shown in Figure 2, which may impede the relevance of the estimated sea surface conditions. To address this issue, we propose to combine this observation model with a learning-based variational formulation as introduced in the next section.

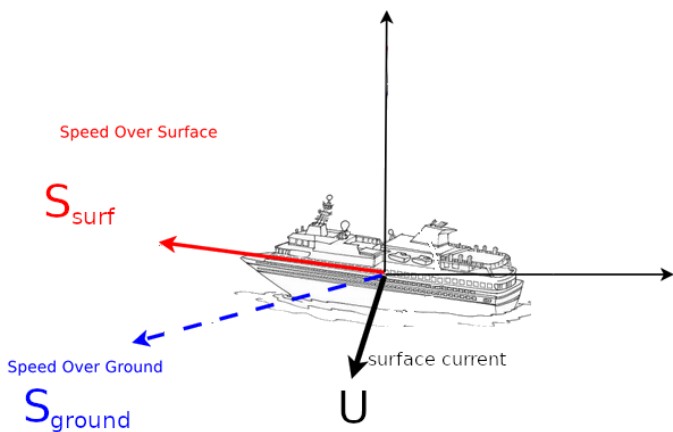

**Figure 1. Observational model:** Under ideal condition, the motion of a boat over the ground is modeled using a linear relation: *speed over ground = Speed on surface + current*. This assumption is no more valid in presence of external forcing (for example in stormy conditions).

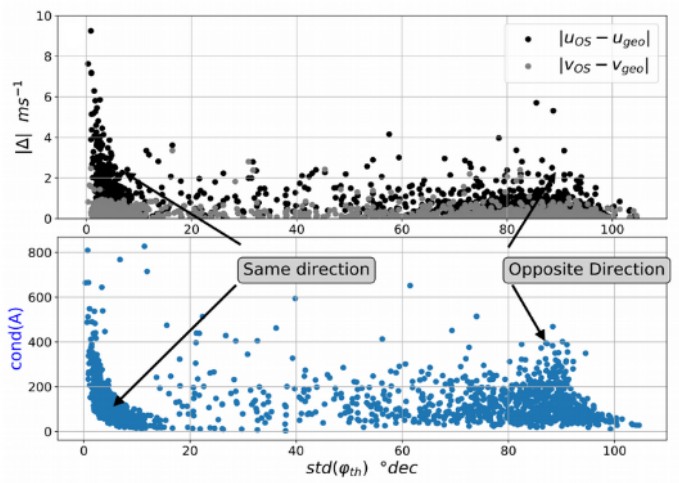

**Figure 2. Reliability of the considered linear model observation:** the validity of modeling assumption depends on the diversity of the observed boat trajectories. We first report a comparison between the absolute error between the AIS-derived and altimetry-derived currents (**top**). We also depict the conditioning of the linear system associated to the considered observation model (**bottom**). The greater the inconsistency between AIS-derived and altimetry-derived currents, the worse the conditioning of the linear observation system.

*2.3. Problem Statement*

Given the above-mentioned observation model, the reconstruction of sea surface currents from AIS data streams may be regarded as a data assimilation problem [15]. Formally, this typically leads to solving the minimization of a variational cost

$$X^* = \arg\min_X J(X, Y) + R(X) \tag{2}$$

where $J$ is the data fidelity or observation term, Y the observed data and X the unknown states we aim to estimate. Here, term $J$ is associated with the observation error derived from the linear observation model, the unknown state $X(p,t) := (U(p,t), S_{surf(p,t)})$ and the observational data $Y := (S_{ground}(p,t)$. The regularization term $R(.)$ constrains the inversion such that the minimization of the variational cost becomes a well-posed problem. This regularization term shall exploit some prior knowledge on the considered problem. Geophysical data assimilation approaches typically involve physical models governed by partial differential equations (PDEs). For example in 4D-Var assimilation schemes [7], the prior is generally stated as $R(.) := \sum_{k=1}^{T} \|\mathcal{M}(x_{k-1}) - x_k\|$ where $\mathcal{M}$ denotes a forecast operator associated with the numerical integration of a PDE. For the reconstruction of sea surface currents, simplified models derived from Navier–Stokes equations such as quasi-geostrophic models [16] may be appealing. They do however only apply for very specific dynamical upper ocean regime, which question their applicability in our case.

Optimal interpolation (OI) [17,18] relies on covariance-based prior for regularization term $R(.)$ in variational formulation (2). Combined with linear-quadratic observation model, OI comes to compute the analytical solution minimization (2). OI is widely exploited for operational and state-of-the-art geophysical products, for instance state-of-the-art sea surface currents derived from satellite altimetry data. A key issue is the selection of the appropriate covariance model [19]. Numerous parametric models have been proposed, including covariance models explicitly related to stochastic PDE priors [20]. In general, due to the underlying Gaussian assumption, OI poorly represent fine-scale processes that depict non-Gaussian features.

Recently, data-driven and learning-based approaches [21–23] have emerged as efficient schemes to solve inverse problems for space–time processes. In this respect, recent works have also introduced trainable variational formulations for inverse problems [24,25]. Such approaches may provide general physics-informed trainable frameworks to exploit some prior knowledge while benefiting from the flexibility and generalization capacities of deep learning methods. As detailed in the next section, we follow this approach to exploit the prior knowledge issued from observation model described by Equation (1) to trainable dynamical priors to represent sea surface dynamics.

## 3. Proposed Approach

In this section, we detail the proposed framework for the learning-based inversion of sea surface currents from AIS data streams. Let us introduce the following notations:

- $Y$ the available set of AIS messages over a time period $[0, T]$;
- $U$ the spatio-temporal sea surface current field we aim to reconstruct;
- $V$ an external dataset of sea surface current fields that we consider realistic such as reanalysis product or numerical simulations.

We state the reconstruction of fields $U$ from observation data $Y$ as the solution of the following optimization problem:

$$U^*, \Theta^* = \arg \min_{U,\Theta} J(U,Y) + \lambda_U R_\Theta(U) + \lambda_V R_\Theta^\star(V) \tag{3}$$

where $J(U,Y)$ is the observation term derived from the linear observation model, $R_\Theta$ a regularization term with trainable parameters $\Theta$, which applies to unknown state $U$ and $R_\Theta^\star$ a regularization term with the same trainable parameters as $R_\Theta$, which applies to external dataset $V$. $\lambda_U$ and $\lambda_V$ are weight parameters balancing the regularization terms w.r.t. the data fidelity term. We may point out that we aim to jointly minimize the above variational cost w.r.t. unknown states $U$ and model parameters $\Theta$. We detail in the subsequent the observation and regularization terms as well as the considered minimization framework.

### 3.1. Observation Term J

Given the observation model defined by Equation (1), we derive as follows the observation term $J$ in the considered variational framework. Using a nearest-neighbor inter-

polation, we merge AIS observations onto a $1/12°$-resolution grid with a 24 h predefined time resolution. This interpolation results in defining an AIS-derived observation tensors $Y \in \mathbb{R}^{(T/\Delta_t) \times N_x \times N_y \times 2}$ with the same space–time resolution as sea surface current variable $U$, where $T$ is the final time, $\Delta_t$ is the time step and $(N_x, N_y)$ refer to spatial dimensions. In the reported experiments, we typically consider a 8-day time window with a daily resolution for $108 \times 72$ fields, i.e., $T = 8$, $\Delta_t = 1$, $N_x = 108$ and $N_y = 72$

The observation cost $J$ is then defined as follows:

$$J(X, Y) = \sum_{t=0}^{T} \|L(U(t), S_{surf}(t), H(t)) - S_{ground}(t)\|_{1,\Omega_t} \qquad (4)$$

where $\Omega_t$ is the observable spatial domain at time $t$. We consider a $L_1$ norm in the observation model to account for possibly noisy and/or corrupted AIS messages [26].

### 3.2. Space–Time Dynamical Prior

We assume that the space–time dynamics of sea surface currents can be approximated by the solution of an ODE on a low-dimensional manifold. Let us denote by $Z$ the low-dimensional latent state, $z_0 \in Z$ an initial condition and trainable operators $f$ and $\Phi$ where $f$ is a Lipschitz function and $\Phi$ a decoder that maps the latent state to the state space containing $U$. The considered state-space formulation is as follows:

$$\begin{cases} U(.,t) &=& \Phi(Z(t)) \\ Z'(t) &=& f(Z(t)) \\ Z(0) &=& Z_0 \end{cases} \qquad (5)$$

The comprehensive modeling of ocean dynamics relies on variables that are not observed here, such as atmospheric forcing or subsurface ocean conditions. In order to take into account these unobserved variables, we consider an augmented latent space as in [27,28]. More specifically, the low-dimensional latent state consists in two components: $Z(t) := [Z_1(t), Z_2(t)]$. While ODE operator $f$ applies to the augmented state $Z$, mapping operator $\Phi$ only makes use of component $Z_1$ for the reconstruction of the sea surface current $U$. As such, the second component $Z_2$ in the latent space encodes unobserved variables and forcings.

We consider a neural implementation of the dynamical system stated in Equation (5). More specifically, we exploit a neural ODE package [29,30] with a fourth-order Runge–Kutta (RK4) integration scheme.

### 3.3. Trainable Regulatization Terms

The considered space–time dynamical prior falls in the category of deep generative models [31]. Variational learning techniques, especially VAE schemes, are among the state-of-the-art methods to train generative models. Broadly speaking, VAEs aim to model the distribution of random variables $x$ using latent variables $z$. The joint distribution of $(x, z)$ is stated as $p(x, z) = p(x|z)p(z)$ where $p(z)$ and $p(x|z)$ are given by some parametric forms. Under this framework, we would typically aim to maximise the likelihood of the observations w.r.t. model parameters. This would lead to the computation of the likelihood of the observed data as

$$p(x) = \int p(x, z) dz. \qquad (6)$$

However this integral is generally intractable. To address this problem, VAE frameworks [32] introduce an inference model $q(x|z)$, which is trained such as $q(x|z)$ approximate the true posterior $p(z|x)$. They exploit a lower bound on the likelihood of the data, referred to as the evidence lower bound (ELBO):

$$\text{ELBO} = E_{q(z|x)}[log\,p(x|z)] - KL(q(z|x)||p(z)) \qquad (7)$$

where $E_{q(z|x)}$ refers to the expected value under density probability $q(z|x)$ and $KL$ to the Kullback–Leiber divergence. One may then use the negative ELBO as training loss. VAEs show experimentally excellent capabilities of learning low-dimensional representations of high-dimensional variables. They may result in highly structured spaces which linearize small deformations [33]. This provides the theoretical basis for considering ELBO-based regularization terms in Equation (3). In this work we suppose $p(x|z) \sim \mathcal{N}(\Phi(z), I)$ where $I$ denotes the identity matrix, which is a common practice when training VAEs.

More precisely, given the space–time dynamical prior introduced in the previous section, we define regularization term $R_\Theta^\star(U)$ according to the following ELBO formulation:

$$R_\Theta^\star(U) = \sum_{t=0}^{T} KL(\Psi(U_t)||\mathcal{N}(0,1)) + \|U_t - \Phi(\Psi(U_t))\|_2^2 \qquad (8)$$

where $\Psi$ is a trainable mapping operator from spatio-temporal space to the latent space, which relates to the parameterization of approximate posterior $q()$. As classically applied in VAE settings, we considered a zero-mean and unit-variance normal prior, in the latent space, denoted as $\mathcal{N}(0,1)$.

As suggested in [28], we also consider a temporal regularization term in the latent space:

$$R_\Theta(U) = R_\Theta^\star(U) + \sum_{t=0}^{T-\Delta_t} \|RKI(Z(t), f) - Z(t+\Delta)\|_2 \qquad (9)$$

where $RKI$ is a one-step-ahead RK4 integration scheme that forecasts the latent state at time $t + \Delta$ given any initial condition $Z(t)$ at time $t$. This term enforces the time consistency of the inferred latent states according to the trained dynamical operator $f$.

### 3.4. Training and Evaluation Phase

Given variational formulation Equation (2) and the neural-network-based components introduced in the previous sections, we end up with an end-to-end learning framework. Given some AIS dataset $Y$, the training phase aims to solve minimization Equation (3) w.r.t. both sea surface currents $U$ and all model parameters $\Theta$.

For the training phase, we may emphasize that Equation (3) includes a regularization term $R_\Theta^\star(V)$ computed over a representative dataset of exemplars of sea surface current fields. As AIS data streams involve an irregular and sparse space–time sampling of the space–time domain conditioned by maritime traffic routes, such exemplars shall provide additional information to constrain the space–time variability encoded by the latent ODE representation. In our implementation for minimization (3), we use Adam optimizer for model parameters and a fixed step gradient descent for retrieving initial conditions of Equation (5).

Given a trained model with parameters $\Theta$, we may solve minimization (2) to reconstruct the sea surface currents associated with any new AIS dataset. Similarly to the training phase, we consider a fixed-step gradient descent optimizer. We may point out that in this case term $R_\Theta^\star(V)$ may be dropped from minimization (3).

## 4. Application to a Real AIS Dataset

### 4.1. Experimental Setting

As case study region, we consider a region off South Africa, which involves a dense maritime traffic. This area associated with the Aghulas current exhibits complex upper ocean dynamics and is considered as a challenging region for current estimation using satellite altimetry [34]. The collected AIS dataset covers 240 days in 2016 with more than 4 millions of AIS messages. Figure 3 provides an illustration of the daily sampling pattern of the AIS data.

To account for the seasonal variability of upper ocean dynamics in the case study area, we apply and evaluate the proposed approach to three different distinct periods of 80 days, namely

- a summer period from 1 January to 20 March;
- a transition period from 9 April to 28 June;
- a winter period from 29 June to 16 September.

We consider two state-of-the-art operational products for benchmarking purposes and as external complementary dataset of numerical examplars *V* during the training phase: namely, GLORYS (the GLORYS reanalysis product used for the experiment is available at https://marine.copernicus.eu/ (accessed on 6 May 2021)) reanalysis product (ref. [9] and altimetry-derived AVISO product (we refer the reader to the following link for additional information on altimeter-derived AVISO sea surface currents. https://www.aviso.altimetry.fr/en/data/products/sea-surface-height-products.html (accessed on 6 May 2021)). GLORYS product relies on the variational assimilation of an ocean circulation model based on coupled Navier–Stokes equations, using different sources of observation data especially sea surface temperature (SST) and altimeter-derived sea level anomaly (SLA). This dataset results in a 22-year time series from 1993 to 2016. By contrast, AVISO product is a purely observation-driven dataset based on the analysis of altimeter data from multiple satellite missions using optimal interpolation. For benchmarking purposes, besides AVISO and GLORYS sea surface currents, we also include a AIS-derived product combining an expert-based processing and optimal interpolation (OI) [2], referred to as AIS-OI.

Hereafter, we refer to the different configurations of proposed approach as VAE-NODE-AIS, VAE-NODE-AIS-GLORYS and VAE-NODE-AIS-AVISO.

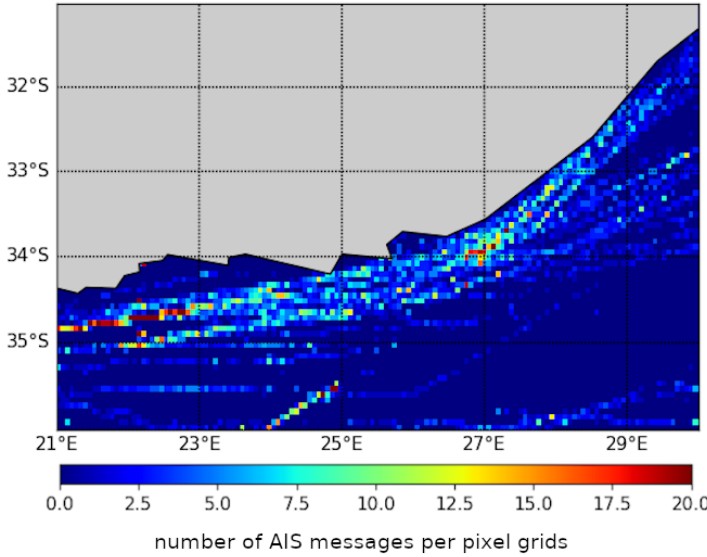

**Figure 3. Example of daily coverage of the AIS data available on a daily basis:** here 16 January 2016, for the considered case-study area, off South Africa.

Regarding the evaluation framework, we exploit an independent in situ datasets issued from drifters as depicted on Figure 4 to evaluate the relevance of the reconstructed sea surface currents. As performance metrics, we focus on the mean square error (MSE) of the estimated sea surface current along drifters' trajectories. Overall, the in situ validation datasets comprise respectively 722, 4977 and 752 samples for each of the three evaluation periods.

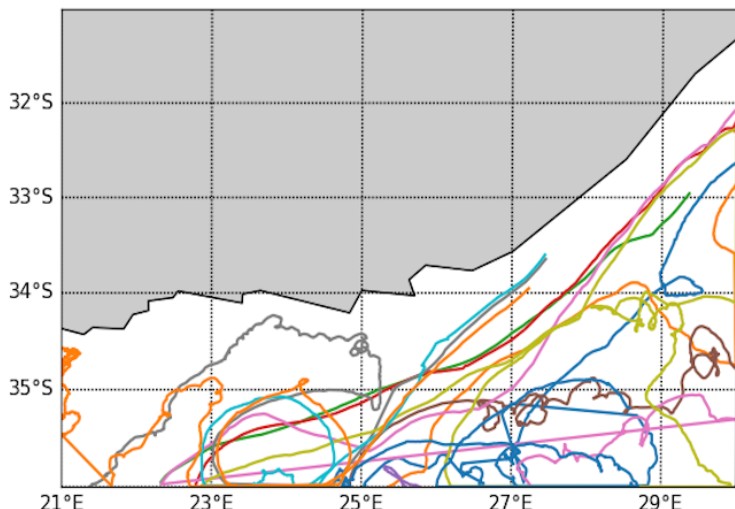

**Figure 4. Trajectory of drifters used for performance evaluation:** Over the three considered period. This dataset contains $N = 6451$ points of measurement of current velocity $u_i$ at position $p_i$ and time $t_i$. For benchmarking purpose, we compare different current product using mean square error (MSE) where $MSE(U) := \frac{1}{N} \sum_{i=1}^{N} \|U(p_i, t_i) - u_i\|_2^2$.

### 4.2. Neural Network Architecture

To implement the proposed end-to-end learning framework, we state a specific neural network architecture for the proposed space–time dynamical prior. It consists in the parameterization of encoder $\Psi$, decoder $\Phi$ and ODE operator $f$:

- Encoder $\Psi$ applies a succession of four convolution layers with ReLU activations, followed by three dense layers with ReLU activation;
- ODE operator $f$ is parametrized by five dense layers with Soft ReLU activation, which guarantees $f$ to be Lipschitz;
- Decoder $\Phi$ exploits only the component $Z_1$ of the latent variable $Z$. It involves a four-layer ConvTranspose network with ReLU activation. The output is stated as $\Phi(Z) = \lambda tanh(h_k)$ where $h_k$ denotes the value of the last ConvTranspose output. For all the experiments, we set $\lambda := 8$.

Figure 5 depicts the resulting architecture.

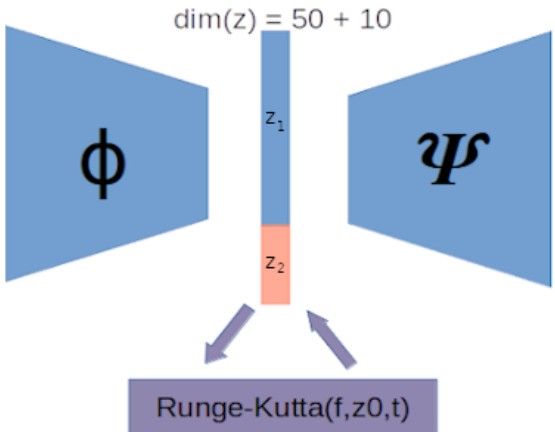

**Figure 5. Neural-network parameterization of the considered space–time dynamical prior:** it comprises a CNN-based encoder $\Psi$, a CNN-based decoder $\Phi$ and an ODE model $f$. Current grid U is expressed as $U := \Phi(z) = \Phi(z_1)$ where $z = [z_1, z_2]$ follow the ordinary differential equation $z'(t) = f(z(t))$ with initial condition $z(0) = z_0$.

*4.3. Results*

The results obtained from independent in situ data are reported in Table 1 (lowest MSE are in bold). The proposed method is denoted as VAE-NODE and trained either using solely AIS data or using both AIS data and external datasets (GLORYS, AVISO and OI).

**Table 1.** Reconstruction performance evaluate from independent in situ data.

| Data | Method | Summer (MSE) | Autumn (MSE) | Winter (MSE) |
|------|--------|--------------|--------------|--------------|
| SLA | AVISO | 0.1154 | 0.0802 | **0.0377** |
| SLA +SST | GLORYS | 0.0687 | 0.0816 | 0.0422 |
| AIS | OI | 0.0855 | 0.1330 | 0.1885 |
| AIS | VAE-NODE | 0.1401 | 0.1458 | 0.1563 |
| AIS + OI | VAE-NODE | 0.4001 | 0.1809 | 0.2176 |
| AIS + AVISO | VAE-NODE | 0.1469 | 0.1222 | 0.1127 |
| AIS + GLORYS | VAE-NODE | **0.0498** | **0.0738** | 0.0378 |

Among the different configurations of the proposed VAE-NODE framework, we report the best performance for all periods using a training with GLORYS data as external dataset *V* in Equation (3). It may be noted that using AIS alone or using AVISO data as external dataset significantly lessens the performance (e.g., 0.05 for VAE-NODE-AIS+GLORYS setting vs. 0.40 (resp. 0.14) for VAE-NODE-AIS+AVISO (resp. VAE-NODE-AIS) setting for the summer period). The mean reconstruction performance is also significantly better than that of AIS-OI scheme.

Interestingly, the proposed approach outperforms GLORYS and AVISO products for summer and autumn periods, with a significant relative gain, up to 28% w.r.t. GLORYS products and 57% w.r.t. AVISO. For the winter period, we reach a reconstruction performance similar to AVISO product and slightly better than GLORYS product.

We provide an illustration of the reconstructed velocity fields on 16 January 2016 in Figure 6. The sea surface current estimated by VAE-NODE-AIS+GLORYS depicts information at fine scales (e.g., the width of the current coming from the north-east of the area) and sharp gradients compared with the altimeter-derived AVISO method. Whereas the AIS-OI baseline involves local artifacts, which relate to noise patterns of the AIS dataset and local outliers for simplified observation model (1), the proposed framework retrieves more consistent space–time dynamics. This is regarded as a direct outcome of the considered regularization through the proposed latent ODE representation. When using both AIS and GLORYS data in the training phase, we retrieve much larger velocity values along the high-current area. As the VAE-NODE-AIS and VAE-NODE-AIS+GLORYS configurations reach similar values for training loss *J*, this suggests that the additional use of GLORYS data greatly improves the interpolation capabilities of the trainable latent ODE representation.

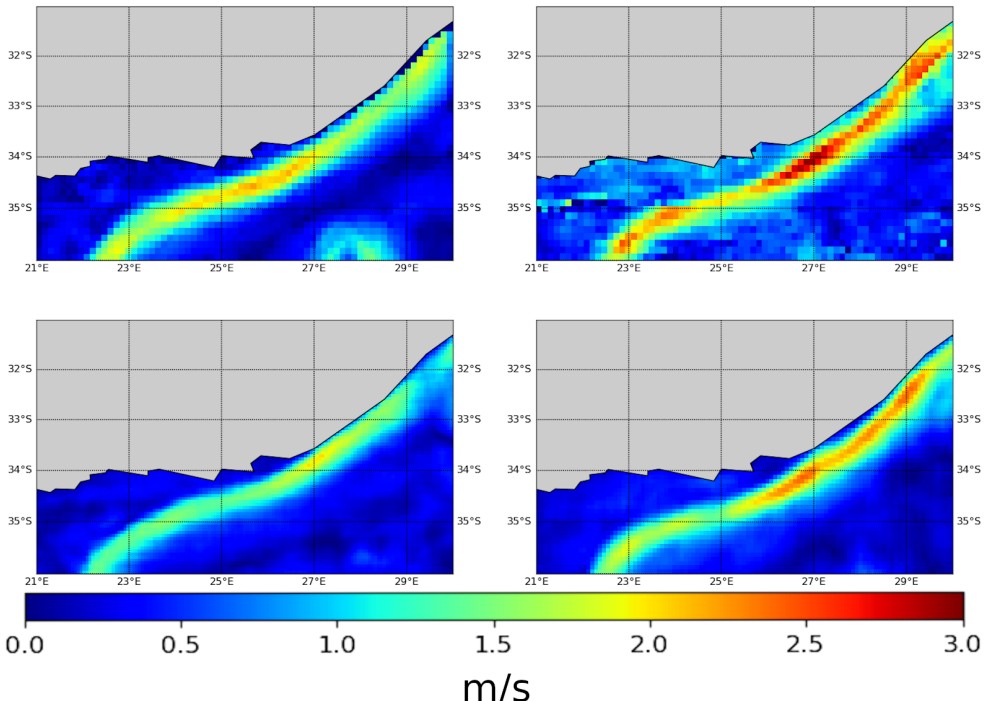

**Figure 6. Reconstructed velocity fields for 16 January 2016: Top left**: AVISO, **top right**: AIS-OI, **bottom left** proposed scheme using only AIS data during the training phase (VAE-NODE-AIS), **bottom right**: proposed scheme using AIS and GLORY during the training phase (VAE-NODE-AIS+GLORYS).

*4.4. Ageostrophy*

Altimetry-derived current relies on a geostrophic assumption [35]:

$$u_g := -\frac{g}{f} \nabla \times h \tag{10}$$

where $f$ denotes the Coriolis parameter and $g$ the gravitational acceleration and $h$ a surface over the geoid computed using sea level anomaly (SLA) and the mean dynamic topography (MDT). Therefore altimetry-derived currents are divergent-free and cannot account for the divergence of the total sea surface currents, which relate among others to wind-induced and vertical exchange processes.

By contrast AIS-derived currents involve no such geostrophic assumptions. We explore here the extent to which the reconstructed AIS-derived sea surface dynamics involve some non-null divergence patterns.

We report in Figure 7 the mean divergence map of the reconstructed fields over the processed 240-day period. Both AIS-OI and VAE-NODE-AIS+GLORYS involve much more contrasted patterns with strongly negative and strongly positive divergence values compared with GLORYS product. This result is particularly interesting for VAE-NODE-AIS+GLORYS as it also comes with much lower MSE for the estimated velocities along available drifters' trajectories.

Estimating the real divergence is of great importance in oceanography because it informs on the vertical velocities inside the upper layers of the ocean. Positive divergence values relate positive vertical velocities often associated with up-welling areas. It is clearly the case in the southwestern zone of the case-study region close to the shore located in the north of the Agulhas current. This area of positive divergence is known to be rich in nutrient as a result of intense up-wellings [36], which further supports the relevance of the proposed scheme.

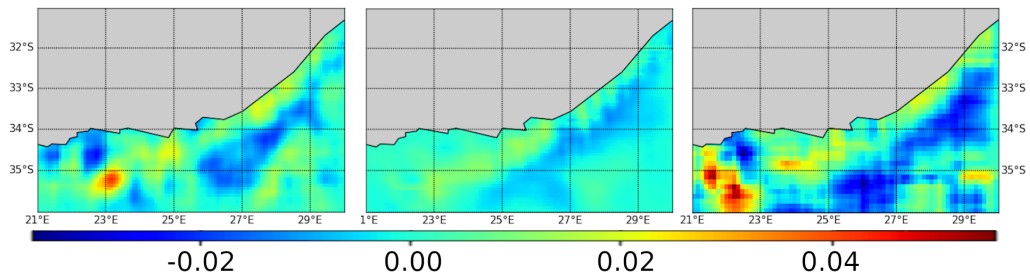

**Figure 7. Mean divergence of the reconstructed velocity fields:** AIS-derived current allow to retrieve divergent part of surface current. Here we compute the divergence of different current product over the first summer evaluation period. **Left**: proposed approach using AIS and GLORYS data during training (VAE-NODE-AIS+GLORYS). **Middle**: GLORYS reanalysis product. **Right**: the AIS-OI baseline method.

*4.5. Generalization Capacity*

We also investigate the generalization capacity of the trained neural network models: To this end, we apply the model trained for a given period to the two other ones, such that only optimize the variational cost w.r.t. initial conditions in the latent space by gradient descent. The associated optimization problem (Equation (11)) is of much lower dimension than the original one because we only explore initial conditions associated to the dynamical model on the latent space.

$$U^* = \arg\min_Z J(U, Y) + \lambda_U R_\Theta(U) \tag{11}$$

In these experiments, we consider VAE-NODE-AIS+GLORYS models. We report results in Table 2.

**Table 2.** Generalization capabilities of the trained neural network models.

| Method | Summer (MSE) | Autumn (MSE) | Winter (MSE) |
|--------|--------------|--------------|--------------|
| summer | **0.0498** | 0.1104 | 0.0760 |
| autumn | 0.1101 | **0.0738** | **0.0367** |
| winter | 0.1275 | 0.1210 | 0.0378 |

Whereas the models trained for the summer and winter periods do not seem to apply very relevantly to the other periods with a significantly lower reconstruction performance, the model trained for the autumn period reaches the best reconstruction performance along drifters' trajectories both for autumn and winter periods. These results support the training of season-specific models to account for season-related upper ocean dynamical regimes.

*4.6. Sensitivity Analysis*

We further evaluate the sensitivity of the reconstruction performance of the proposed framework w.r.t. key parameters, namely:

- the size of the latent space (Table 3);
- the size $T$ of the time window of integration of the dynamical systems (Table 4).

We perform this sensitivity analysis for the summer period using the MSE for the drifters' data as performance metrics.

**Table 3.** Sensitivity analysis: latent space dimension. We report the reconstruction performance depending on the dimension of the latent space. Each configuration "$a + b$" refers to a $a$-dimensional output $Z_1$ of the encoder $\Psi$ and a b-dimensional augmented component $Z_2$ to take account for unobserved dynamics.

| Latent Space Dim | MSE |
|---|---|
| 50 + 0 | 0.0779 |
| 50 + 10 | 0.0498 |
| 50 + 20 | 0.0491 |
| 50 + 50 | 0.0620 |
| 10 + 10 | 0.0805 |
| 250 + 10 | 0.0621 |

Table 3 shows that the use of an augmented state clearly brings some improvement. For instance, for configuration "50 + 10", i.e., a 10-dimensional augmented component, we report an improvement of 35% in Table 3 compared to the configuration "50 + 0" without an augmented latent space. When considering a higher-dimensional augmented component, the improvement seems marginal, as illustrated by a relative gain of 1.5% when considering 20-dimensional augmented component rather than a 10-dimensional one, the 50-dimensional component configuration lead a lower variational cost, but we lose 25% on the performance evaluation. This suggest that the dynamical model overfit the observations. The dimension of the output of encoder $\Psi$ also significantly affects the reconstruction performance. A 50-dimensional parameterization led to the best results. For lower-dimensional encoder output, the representation issued from the encoder may not be informative enough, which leads to higher variational cost and a lower reconstruction performance w.r.t. drifters' trajectories. For a higher-dimensional encoding space, the training leads to a variational cost close to that observed with the reference configuration but the model performance lessen by 25% in terms of reconstruction of sea surface currents along drifters' trajectories.

**Table 4.** Sensitivity analysis: impact of the width of the integration time window.

| $\Delta T$ | MSE |
|---|---|
| 4 days | 0.0625 |
| 8 days | 0.0498 |
| 16 days | 0.1231 |

Table 4 shows the effect of the parameters $\Delta T$ on the reconstruction performance. As expected, when $\Delta T$ is too small, typically below 96h, the identification of the sea surface current field may be more unstable due to the scarsity of available observation data whereas for a high value of $\Delta T$, typically above 384, the resulting time regularization through the ODE prior may be too strong and may negatively impact the reconstruction performance.

## 5. Discussion

In this work, we investigate the use of AIS maritime data and deep learning methods for the reconstruction of sea surface current. Within a 4D-Var assimilation framework, our deep learning approach combines a geophysically sound observation model with a trainable physics-informed prior. Numerical experiments show that the exploitation of reanalysis data provides relevant information to better constrain the training of the considered space–time prior onto sea surface dynamics stated as a reduced-order model [37]. This training strategy results in a significant gain in terms of sea surface current reconstruction as well as in terms of generalization capabilities.

Through numerical experiments on a real dataset, we show that AIS data in a case-study region with a dense maritime traffic constitute a reliable source of observations to

derive sea surface currents, which may better match in situ measurements than state-of-the-art satellite-derived and assimilation-based products. Interestingly, as AIS-derived sea surface currents do not rely on any geostrophy assumption, they can reveal the rotational-free ageostrophic component of the sea surface dynamics, which cannot be informed by satellite altimetry and remains under-estimated by assimilation-based products. As such, the proposed framework could contribute better assess from in situ measurements the space–time variabilities of ageostrophic flows.

From a methodological point of view, the proposed framework also opens new research avenues to exploit the synergies between different observation datasets. While, here, our learning-based inversion only relies on AIS data, we could complement the considered variational model with other observation terms. In this line of work, the synergy between altimetry and AIS data seems particularly appealing to better exploit these two data sources that may significantly complement each other in terms of space–time sampling and dynamical features they can reveal. Future work shall also investigate extensions of the proposed approach to the inversion of other geophysical processes, such as sea surface waves and winds, which also affect ship trajectories and may then be encoded in AIS data streams.

**Author Contributions:** Investigation, S.B.; Methodology, S.B. and R.F.; Resources, Y.G.; Supervision, C.L. and R.F.; Validation, C.L.; Writing—original draft, S.B.; Writing—review & editing, F.R. All authors have read and agreed to the published version of the manuscript.

**Funding:** This research received no external funding.

**Institutional Review Board Statement:** Not applicable.

**Informed Consent Statement:** Not applicable.

**Data Availability Statement:** Not applicable.

**Acknowledgments:** This work was supported by Eodyn, LEFE program (LEFE MANU project IA-OAC), CNES (grant SWOT-DIEGO) and ANR Projects Melody and OceaniX. It benefited from HPC and GPU resources from Azure (Microsoft EU Ocean awards) and from GENCI-IDRIS (Grant 2020-101030).

**Conflicts of Interest:** The authors declare no conflict of interest.

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
