# Peer review of "Unsupervised Reconstruction of Sea Surface Currents from AIS Maritime Traffic Data Using Trainable Variational Models"

_remotesensing, doi:10.3390/rs13163162_

Round 1

Reviewer 1 Report

Main concern

The idea to extract many other info from the AIS data is very interesting, but how to confirm the reliability of the results has to be significantly improved.

For instance, why do not compare your results with other independent data, like oceanographic data on the currents in the same are of the case study?

Other issues

1) some references are missed, for example at rows 28-29 or with question marks (e.g., rows 114-115, 190, 192 par 2.3)

2) paragraph "problem statement" repeated, renumber the formulas accordingly

3) figure 5 is missed

Author Response

We thank you for taking time to review this paper and for your valuable comments and suggestions, please find below our answers :

« why do not compare your results with other independent data, like oceanographic data on the currents in the same are of the case study? »

In this work we use a dataset of drifters and mean square error (MSE) as a metric for performance evaluation. For benchmarking purpose, we consider two altimeter-derived product refered as AVISO and GLORYS. Drifters measurement constitute the only source of datas that could be considered as ground-truth measurement. There exist others oceanic tracers such as sea surface temperature (SST) or chlorophyll density fields that could be correlated to sea-surface current but among our knowledge, we doesn’t know physical model that could perfectly associate this tracers to sea-surface current.

In the proposed approach, we make use of the assimilated product GLORYS as a dataset of realistic representations but the operator that link AIS datas to sea-surface current is generic (it is stated as the integration of a gradient flow). If a dataset of ground-truth observed representation would exist, we think that using a supervised learning framework to associate sea-surface current to AIS datas would lead to better results, and the proposed methodology could have been quite different for this data assimilation task.

Reviewer 2 Report

This paper proposed an unsupervised reconstruction of sea surface currents from AIS maritime traffic data using trainable variational models. This method can resolve spatial scales below ~100km. It is well-written, and easy to follow. However, some critical issues should be classified by a Major Revision.

Major comments:

Is it the first time that the authors use AIS to handle such task, even to ~100km?

Minor comments:

Line 14, maritime traffic is very important for MDA, so the main participant is ship, e.g., some work from high-speed ship detection in sar images based on a grid convolutional neural network, and depthwise separable convolution neural network for high-speed sar ship detection. The authors should add these introductions to expand the true importance of their work.

How to obtain suitable ASI information, how to interpolate time when the AIS is lost, e.g., the work from ls-ssdd-v1.0: a deep learning dataset dedicated to small ship detection from large-scale sentinel-1 sar images. This is challenging as in Line 32-33.

Line 49, GPU acceleration? Which type of GPU? 2080Ti from hyperli-net: a hyper-light deep learning network for high-accurate and high-speed ship detection from synthetic aperture radar imagery, and balance scene learning mechanism for offshore and inshore ship detection in sar images.

Another problem, are you focusing on the ocean currents of major transportation routes? Namely, consider ships?

Figure 1, estimate the speed of ships?

Please give a detailed introduction of AIS information you obtained.

Line 192, the below, [?] ?

Why use the KL?

The methodology is not clear at all. Please give the flow chart and block diagram.

What about the GPU time and CPU time, e.g., the reports of shipdenet-20: an only 20 convolution layers and <1-mb lightweight sar ship detector.

To sum up, please the authors must address the above problem, seriously and more seriously. Otherwise, it cannot meet the publishing standards of the remote sensing journal.

Author Response

We thank you for taking time to review this paper and for your valuable comments and suggestions. please find our answers below :

« Is it the first time that the authors use AIS to handle such task, even to ~100km? »

It is not, we can cite : « Oceanic Circulation in the Strait of Gibraltar revealed by AIS data Information » C. Legoff et al. But this is to our knowledge the first deep-learning based approach to adress the regularization of the ill-posed inverse problem associated to AIS datas.

« How to obtain suitable ASI information, how to interpolate time when the AIS is lost, e.g., the work from ls-ssdd-v1.0: a deep learning dataset dedicated to small ship detection from large-scale sentinel-1 sar images. This is challenging as in Line 32-33. »

The AIS dataset were provided by a private provider, in fact we doesn’t perform any filtering on the available dataset, except we consider only AIS messages that contains all the informations used by the linear observational model (the heading, the speed overground, and the course overground).

« Line 49, GPU acceleration? Which type of GPU? 2080Ti from hyperli-net: a hyper-light deep learning network for high-accurate and high-speed ship detection from synthetic aperture radar imagery, and balance scene learning mechanism for offshore and inshore ship detection in sar images. », « What about the GPU time and CPU time, e.g., the reports of shipdenet-20: an only 20 convolution layers and <1-mb lightweight sar ship detector. »

Training time on a single GTX 1080 ti is typically 24 hours.

« Figure 1, estimate the speed of ships? »

Yes, For each observed pixel and under the linear observational model of sea-surface current associated to AIS datas, we need to perform a minimization which involves the surface speed of ships (which is unknwon) and the value of sea surface current at the considered pixel grid. AIS datas only provides speed of ships over ground. (GPS information).

« Why use the KL? »

In this works we choose the variational autoencoder framework (VAE) to perform spatial regularization of sea-surface current. We think that the VAEs constitute an appealing framework for data assimilation tasks due to their ability to linearize small deformations (for interpolation purpose for example) and learning underlying probability density associated to geophysical fields. In the VAE framework, the Kullbach-Leiber divergence (KL) term is associated to the estimated lower bound (ELBO) loss which is a lower bound on probability density associated to datas. Training a VAE consist in maximizing this lower bound which include a reconstructioin loss and this KL term.

The KL terms acts as a regularization term over the distribution of latent vectors. Here we choose to enforce latent vectors to follow a gaussian distributions which is a common practice when training a VAE.

« Another problem, are you focusing on the ocean currents of major transportation routes? Namely, consider ships? »

The proposed approach relies on AIS messages provided by ships, there exist some ocean region without maritime trafic, this means that for these area we can not use AIS datas for the reconstruction of sea surface current at fine scale. To have a sketch of idea of the global coverage of AIS data streams in real time, please find in attached file a screeenshot from the website https://www.marinetraffic.com.

We thanks you for the proposed additional references.

Reviewer 3 Report

The authors present interesting paper about their investigation which is about the use of AIS maritime data and deep learning methods for the reconstruction of sea surface current. . More clearly results. All figures with charts could be bigger. Very good job

Author Response

We thanks you for taking time to review this paper, we will update our figures in the revised version of the paper.

Reviewer 4 Report

This is an interesting approach to reconstruct sea surface current. Although the model is subject to further improvement (particularly with wind and waves), the manuscript could be acceptable as the study will potentially impact on certain groups of people.

Minor suggestions:

  • Eq1 – Define U (although it is obvious)
  • Line 160 – Check spelling
  • Eq7 – Define E & KL
  • Line 165 – Define In
  • 4 – Indicate equation numbers properly
  • Based on the text, Figure 4 should come before Figure 3
  • Figure 3 - Velocity map would be informative to compare with Figure 6
  • Figure 4, 6 & 7 – Provide units for color bars
  • Table 1 & 2 – What are these values?
  • Figure 7 – In caption, replace top & bottom with left & right
  • Explain section 3.5 a bit more (how to calculate the capabilities, how to read Table 2, etc.). Mention Table 2 somewhere in text.
  • Line 306-309 – This part needs to be in the caption of Table 3
  • Finally, grammatical errors need to be checked.

Author Response

We thank you for taking time to review this paper and for your valuable comments. We will update our manuscript by taking account of your suggestions and remarks.

Round 2

Reviewer 2 Report

Thanks for the authors' efforts to revise the paper. I think it is a good shape for publishing. Accept!

This manuscript is a resubmission of an earlier submission. The following is a list of the peer review reports and author responses from that submission.